# Assessing the Clinical Characteristics and Management of COVID-19 among Pediatric Patients in Ghana: Findings and Implications

**DOI:** 10.3390/antibiotics12020283

**Published:** 2023-02-01

**Authors:** Israel Abebrese Sefah, Seth Adade Sarkodie, Giuseppe Pichierri, Natalie Schellack, Brian Godman

**Affiliations:** 1Pharmacy Practice Department, School of Pharmacy, University of Health and Allied Sciences, Ho PMB 31, Ghana; 2School of Pharmacy, University of Health and Allied Sciences, Ho PMB 31, Ghana; 3Microbiology Department, Torbay and South Devon NHS Foundation Trust, Lowes Bridge Torbay Hospital, Torquay TQ2 7AA, UK; 4Department of Pharmacology, Faculty of Health Sciences, University of Pretoria, Pretoria 0084, South Africa; 5Department of Pharmacoepidemiology, Strathclyde Institute of Pharmacy and Biomedical Sciences, University of Strathclyde, Glasgow G4 0RE, UK; 6Department of Public Health Pharmacy and Management, School of Pharmacy, Sefako Makgatho Health Sciences University, Pretoria 0208, South Africa; 7Centre of Medical and Bio-Allied Health Sciences Research, Ajman University, Ajman 346, United Arab Emirates

**Keywords:** antimicrobials, children, COVID-19, Ghana, guidelines, hospitals, outcomes

## Abstract

There is an increasing focus across countries on researching the management of children admitted to hospital with COVID-19. This stems from an increasing prevalence due to new variants, combined with concerns with the overuse of antimicrobials driving up resistance rates. Standard treatment guidelines (STGs) have been produced in Ghana to improve their care. Consequently, there is a need to document the clinical characteristics of children diagnosed and admitted with COVID-19 to our hospital in Ghana, factors influencing compliance to the STG and treatment outcomes. In all, 201 patients were surveyed between March 2020 and December 2021, with males accounting for 51.7% of surveyed children. Those aged between 6 and 10 years were the largest group (44.8%). Nasal congestion and fever were some of the commonest presenting complaints, while pneumonia was the commonest (80.6%) COVID-19 complication. In all, 80.0% of all admissions were discharged with no untreated complications, with a 10.9% mortality rate. A combination of azithromycin and hydroxychloroquine (41.29%) was the most prescribed antimicrobial regimen. Compliance to the STG was variable (68.2% compliance). Increased compliance was associated with a sore throat as a presenting symptom. Mortality increased following transfer to the ICU. However, current recommendations to prescribe antimicrobials without demonstrable bacterial or fungal infections needs changing to reduce future resistance. These are areas to address in the future.

## 1. Introduction

The novel coronavirus disease (COVID-19) was first reported in China on the 29 December 2019, and later developed into a pandemic affecting all countries globally [1]. The disease, caused by infection with severe acute respiratory syndrome coronavirus 2 (SARS-CoV-2) spread swiftly from China and Asia across all countries [1]. The first case of COVID-19 in Africa was reported in Egypt in February 2020, with other African countries, including Ghana, following there after [2,3,4]. As of 1 July 2020, there were over 10 million confirmed cases globally with just over 500,000 recorded deaths [5]. 

The first two confirmed cases of COVID-19 in Ghana were reported on 12 March 2020 [4], and by the end of June 2020 there were 17,741 confirmed cases with 112 deaths [3]. Initially, prevalence and mortality rates were appreciably lower among African countries during the early stages of the pandemic compared with, for instance, Western European countries. This was helped by the earlier introduction of lockdown measures and other initiatives as well as younger populations with lower incidence rates of the infection [3,4], with lockdown and other public health measures proven to be effective in slowing down the spread and the impact of the virus in the absence of effective treatments and vaccines [5,6]. The standard of care in Ghana during the pandemic required that suspected persons are quarantined to reduce transmission, while confirmed cases are transferred to designated treatment centers with effective isolation and disease control capacity. Critically ill patients should subsequently be admitted to an intensive care unit for further management [7]. 

The greatest concern across countries, including among African countries, with the rapid implementation of lockdown and other measures, has been the unintended consequences on the management of non-communicable diseases as well as vaccination programs for children [8,9]. The lack of routine vaccinations in children in Africa has been an appreciable concern with some authors predicting that for every one excess death from COVID-19 that could occur among parents and others when children visit vaccination clinics, 84 deaths could be prevented by sustaining routine childhood immunization programs [9,10]. 

Another key concern has been the rapid adoption of re-purposed medicines, including hydroxychloroquine, anti-viral medicines such as remdesivir, as well as anti-parasitic medicines including ivermectin for use in many African countries, despite no robust evidence supporting their use [11,12,13,14,15]. This adoption has been fueled by misinformation on social media and other platforms [11]. The 2020 Ghana Standard Treatment Guidelines (STGs) for COVID-19 treatment, which is yet to be updated, recommends the use of hydroxychloroquine and chloroquine phosphate for the management of all confirmed asymptomatic and symptomatic COVID-19 cases using real time-polymerase chain reaction tests [7]. In addition, azithromycin is empirically indicated for all mild to moderate cases of COVID-19 in addition to hydroxychloroquine [7]. The increased use of these recommended antimicrobials is a concern as this may potentially increase adverse reactions, antimicrobial resistance (AMR), costs and mortality without any additional evidenced-based benefits to patients [13,14,16]. The only exception to the use of re-purposed pharmacological agents in COVID-19 patients has been dexamethasone, which has proven to help hospitalized patients needing ventilatory support [17]. Alongside this, there is growing anxiety with excessive prescribing of antimicrobials in patients with COVID-19, including children, across countries despite limited evidence of bacterial or fungal co-infections [18,19,20,21,22,23]. These high rates of antimicrobial prescriptions may be due to symptoms including coughs and fever which mimic some bacterial infections, as well as concerns with secondary infections especially if this increases mortality [23,24]. However, as mentioned, there are considerable concerns that the overprescribing of antimicrobials will increase AMR [24,25,26,27]. This issue is particularly important in Ghana, with growing resistance rates [28]. 

Children typically have lower infection rates with COVID-19 as well as milder symptoms compared with adults [29,30,31], with an appreciable number of children being asymptomatic [29,30,31,32]. The typical symptoms of children admitted to hospital with COVID-19 are fever, nausea and respiratory symptoms including coughing, as well as diarrhea, with higher admission rates seen among boys versus girls [20,21,29,30,32]. Between 6% to 10% of children admitted to hospital with COVID-19 experience severe disease, with up to 20% or more admitted to pediatric intensive care units; however, the majority of children may not require ventilatory support [20,21,33]. Since admission rates for children are typically lower than seen in adults, the focus among health authorities for children during the early stages of the pandemic has been on other infectious diseases especially given low vaccination rates following lockdown measures, and the subsequent impact on future morbidity and mortality [9,34].

However, there have been growing concerns with potentially increasing mortality among younger patients with successive waves of the pandemic [35,36,37]. In addition, growing anxiety that children are developing Kawasaki Disease (KD)-like symptoms, now known as Pediatric Inflammatory Multisystem Syndrome (PIMS-TS), alongside experiencing hypoalbuminemia, hyponatremia, leucopenia and respiratory changes with COVID-19 [38,39,40]. These combined symptoms can potentially increase admission to intensive care units (ICUs). This can be an issue in lower- and middle-income countries (LMICs) with limited healthcare resources. It has been reported that whilst a lower percentage of children with severe symptoms are admitted to ICUs in LMICs, deaths among these hospitalized children can be higher [33]. For instance, high mortality rates (40%) were seen at Dr. Cipto Mangunkusumo Hospital, in Indonesia, at the start of the pandemic among children admitted with COVID-19 [41]. However, appreciably lower rates were seen recently among children with COVID-19 admitted to hospitals in Bangladesh (1.4%) and Turkey (0.9%), with nearly all children in hospitals in Bangladesh and India in the early years of the pandemic making a full recovery at 92.5% and 92.2%, respectively, although higher rates were seen in Iran (11% mortality) [20,21,42,43].

With the potential for additional cases of COVID-19 in Ghana, including among children, exacerbated by currently high vaccine hesitancy [44], we believed it was important to document current characteristics and management of children with COVID-19 admitted to our hospital in Ghana. There are also concerns with rising AMR rates in Ghana with the appreciable prescribing of antibiotics among patients hospitalized with COVID-19 across countries [18,19,20,21,22,23]. Consequently, the objective of this study was to document the clinical characteristics of children diagnosed and admitted with COVID-19 to the COVID-19 treatment center of the only quaternary medical and research center in Ghana, factors influencing compliance to current Ghanian STG and treatment outcomes. The intention is to provide guidance to improve the future management of children in this hospital and beyond. Compliance to the current STG is important as there have been concerns with adherence to guidelines for the management of infections in our hospital in the past [45,46,47]. 

## 2. Results

### 2.1. Patient Characteristics

The medical records of 201 pediatric patients diagnosed with COVID-19 from March 2020 to December 2021 were extracted, which represented 9.8% (85/869) of pediatric cases admitted from March onwards in 2020, and 10.4% (116/1119) of cases admitted in 2021.

Male patients accounted for 51.7% of admitted patients with COVID-19 (104/201), while those aged between 6 and 10 years accounted for the majority (44.8%, 90/201) of pediatric patients surveyed with a median age of 9 (6–12) years. Nasal congestion (96.0%, 193/201) and fever (95.0%, 191/201) were some of the commonest presenting symptoms, while pneumonia was the commonest (80.6%, 162/210) COVID-19 complication. There was an overall mortality rate of 10.9% (22/201) among our study population (Table 1). Approximately 70% (140/201) of the COVID-19 pediatric cases had no additional diagnosed co-infection on admission, with only 14.9% (30/201) of the cases having a bacterial co-infection diagnosed.

A combination of azithromycin and hydroxychloroquine (41.29%) was the most prescribed antimicrobial regimen followed by a combination of azithromycin, hydroxychloroquine and remdesivir (18.4%). Azithromycin was the most commonly prescribed antibiotic, either alone (14.4%) or in combination (68.2%), with other antimicrobials including chloroquine, hydroxychloroquine, remdesivir, ceftriaxone, doxycycline, meropenem, clindamycin and amoxicillin (Figure 1).

### 2.2. Bivariate Analysis

The overall compliance to the national COVID-19 STGs based on the choice of antimicrobial prescribed was 68.2%. Compliance to the STG was found to be associated with the reporting of a sore throat, documented in the patient’s medical records as a presenting symptom among the study population (*p*-value = 0.026), and the method of radiological assessment used in diagnosis (*p*-value = 0.05) (Table 1). 

Treatment outcomes were also found to be associated with age category of the patients (*p*-value = 0.020), the duration of their admission (*p*-value = 0.025) and whether they were transferred to ICU (*p*-value = 0.002) (Table 2).

### 2.3. Multivariate Analysis 

Compliance to the Ghanaian COVID-19 STG was independently predicted by a sore throat as a presenting complaint (aOR = 0.39, CI = 0.16–0.94, *p*-value = 0.036) (Table 3), while COVID-19 treatment outcomes were independently predicted by the patients’ transfer to the ICU (aOR = 0.22, CI = 0.07–0.66, *p*-value = 0.007) (Table 4). 

## 3. Discussion

To the best of our knowledge, we believe this study is a novelty among pediatric patients admitted to hospitals with COVID-19 in Ghana.

The calculated incidence rates of COVID-19 among pediatric patients managed at the University of Ghana Medical Centre (UGMC) in 2020 and 2021 were 9.8% (85/869) and 10.4% (116/1119), respectively, with an overall mortality rate of 10.9% (22/201). This mortality rate (10.9%) among pediatric patients diagnosed and admitted for COVID-19 was lower than seen in Bangladesh and Indonesia at the start of the pandemic, at 13.3% and 40%, respectively [41,48]. However, it was higher than a recent study in Bangladesh (1.4%), as well as the rate reported by the Child Rights International in Ghana, which showed that the incidence rate of COVID-19 among the pediatric population was 4.43% of the total population, with only four deaths recorded out of 2180 children with the virus (0.13% of total deaths) by the end of 2020 [20,49]. The higher mortality rate seen in UGMC could be accounted for by the surge of the disease following recent waves with new variants, coupled with a relaxation of restrictions surrounding social contact among the pediatric population following the commencement of the vaccination program [50]. In addition, the high COVID-19 vaccine hesitancy rates currently being reported among the population in Ghana may have contributed to these rates [44]. 

Admission rates among males (104/201, 51.7%) was slightly higher than among females (97/201, 48.3%). This is in line with previous studies conducted in other countries [29,30]; however, this had no association with the two outcomes of interest, i.e., prescribers’ compliance to Ghana COVID-19 STGs concerning the choice of antimicrobials and patient treatment outcomes.

Currently, vaccination against COVID-19 among the pediatric population in Ghana is targeted at those above 15 years. This is a concern since among the pediatric population admitted with COVID-19 at UGMC, those aged from 6 to 10 years formed the majority (44.8%, 90/201) followed by those aged from 11 to 15 years (27.9%, 56/201). This agrees with the Child Rights International report, which showed that majority of the pediatric population who have contracted COVID-19 fell within the age group of 0 to 14 years compared to those aged between 15 and 17 years [49]. It must also be noted that several countries including China, Cuba and the USA, have currently approved COVID-19 vaccination for children below the age of 18 years [51]. In addition, the Pfizer vaccine has now been approved by the US Food and Drug Administration for the vaccination of children between the ages of 5 to 10 years, proving guidance to the authorities in Ghana [51].

The commonly reported symptoms among the admitted pediatric patients included nasal congestion and fever (Table 1). These mirror many reported observational studies; however, a cough was reported less commonly in our study population than reported elsewhere [20,30,33,43,52]. We are not fully sure of the reasons for this, and will be investigating this further. 

Encouragingly, most of the pediatric COVID-19 cases that were admitted in our study were mild to moderate, similar to other studies, with only a limited number of children subsequently admitted to ICUs [20,53,54]. These presentations led mostly to a shorter hospital stay of less than seven days (65.2%, 131/201), with pneumonia (80.6%, 162/201) being the most diagnosed complication followed by acute kidney injury (9.5%, 19/201). Overall, few reported cases (9.5%, 19/201) were transferred to the ICU for more aggressive management, which is also similar to other studies [20]. 

Azithromycin and hydroxychloroquine (41.3%, 83/201) were the most prescribed combination of antimicrobial agents used in the management of COVID-19 among the pediatric patients in our study, followed by the combination of azithromycin, hydroxychloroquine and remdesivir (18.4%, 37/201). A plethora of other antibiotics including meropenem, doxycycline, clindamycin and amoxicillin, were also prescribed, even though just under 15% (Table 1) of the cases had a bacterial coinfection diagnosed, which needs addressing to reduce future resistance rates [27,55]. This is reflected by the fact that compliance to the Ghana COVID-19 STGs concerning the choice of antimicrobials prescribed was sub-optimal, with a reported compliance rate of 68.2% (137/201). However, it must also be emphasized that the recommendations for the use of hydroxychloroquine or chloroquine in both adult and pediatric populations for the management of confirmed COVID-19 asymptomatic cases, and the use of the same antimicrobials in combination with either azithromycin or doxycycline for all forms of symptomatic cases, i.e., from mild to severe symptoms, in the current Ghana STG [7], are at variance with other international guidelines including the WHO COVID-19 guideline, the US National Institute of Health and the UK National Institute of Health and Clinical Excellence (NICE) guidelines. These guidelines recommend against the use of these antimicrobials as there is still no robust evidence for their prescribing, either for SARS-CoV-2 virus treatment or the presumptive treatment of bacteria co-infection [56,57,58]. The increased use of antibiotics either alone, e.g., azithromycin, or in combination with other regimens, without the clinical evidence or suspicion of bacterial co-infection, will, as mentioned, increase AMR and associated morbidity, mortality and healthcare cost without improving patient outcomes [19,26,59]. Consequently, there is an urgent need to review the 2020 Ghana STG for the management of patients with COVID-19 admitted to hospitals, based on the latest evidence. The current situation in Ghana though is similar to a number of other African countries where their STGs routinely recommend the prescribing of an appreciable number of antibiotics when managing patients with COVID-19 in both adults and children [60]. 

COVID-19 diagnosed patients who presented with a sore throat were less likely (OR = 0.39, C.I 0.16–0.94, *p*-value = 0.036) to receive empiric treatment with STG recommended antimicrobials in our study compared to those without this symptom. This may be due to the fact the most commonly associated symptoms of COVID-19 in the pediatric population were fever and nasal congestion, and not a sore throat [30,33,36,52]. Whilst the recovery rate in the hospital was high at approximately nine out of 10 admitted cases, pediatric patients who were transferred to the ICU were less likely (OR = 0.22, C.I 0.07–0.60, *p*-value = 0.007) to be discharged home alive. The higher mortality rate among these critical cases needs to be further investigated to ascertain the availability of essential resources and skills for the effective management of these children to reduce future mortality. We will be following this up in the future.

We are aware of a number of limitations with this study. Firstly, we only conducted the study in one hospital for the reasons stated. Secondly, as this was a retrospective study, some key data sets could not be ascertained and so were not recorded. For instance, we were unable to collect any data on the type of biomarkers used in the screening of potential bacterial co-infections as a way of confirming the presence of suspected co-infections. However, despite these limitations, we believe our findings are robust, providing guidance for the future.

## 4. Materials and Methods

### 4.1. Study Design, Study Site and Population

The study design was a retrospective cross-sectional study of medical records of pediatric patients with laboratory confirmed diagnosis of SARS-CoV-2 infection who were admitted to the COVID Treatment Center at the Medical Department of the University of Ghana Medical Centre (UGMC) between March 2020 and December 2021.

The COVID-19 Treatment Center at UGMC is a quaternary medical and research center located on the campus of the University of Ghana in Accra, Ghana. UGMC is an ultra-modern 1000 bed medical center located within the University of Ghana campus in the capital city of Ghana. The Center operates three major service areas namely Quaternary Level Healthcare provision, Medical Training and Scientific/Medical Research.

In addition to the many departments at the center including obstetrics and gynecology, pediatrics, family medicine, laboratory, pharmacy, emergency medicine, it is one of the foremost primary treatment and referral sites offering specialist care for patients with COVID-19 in Ghana. Consequently, this hospital was chosen for this initial study to research the current management and outcomes of children admitted to hospitals in Ghana with COVID-19.

The medical records of pediatric patients between the ages of 0 and 18 years of age admitted for the treatment of laboratory confirmed SARS-CoV-2 infections during the study period were sourced. The medical records of pediatric patients who had missing data concerning laboratory confirmatory tests were excluded.

### 4.2. Data Collection Method and Analysis

A data collection tool (Appendix A) was adapted from previous studies and designed to extract the medical records of the study participants [42,43]. We have used this approach before in previous studies involving patients with COVID-19 as well as previous point-prevalence studies [20,37,61,62].

Patients’ age, gender, year of admission, the presence of complications, the level of the severity of their COVID-19 spanning from mild to severe symptoms including acute respiratory distress syndrome based on the criteria contained within the current Ghana STG [7], their presenting symptoms, (i.e., fever, cough, sore throat and nasal congestion, treatment outcomes, categorized as discharged home and mortality, their duration of hospital admission, whether the children were transferred to the ICU for subsequent management or not, any antimicrobials prescribed, and compliance of the choice of antimicrobials prescribed based on the level of severity of COVID-19 according to the 2020 edition of the Ghana COVID-19 STG [7]. Adherence to guidelines is increasingly seen as a marker of the quality of care as part of antimicrobial stewardship programs in hospitals [46,61,63,64,65,66].

All data collected were entered on Microsoft Excel version 2013 and imported into STATA version 14 (StataCorp, TX, USA) for analysis. The analysis comprised of descriptive statistics, bivariate analysis (Chi-square test of independence) to determine any association between the study outcomes (guideline compliance and treatment outcome) and the independent variables at a significance level of 95%, and a multiple logistic regression to determine the predictors of the study outcomes, i.e., compliance to guidelines and treatment outcomes.

### 4.3. Ethical Consideration

Ethical clearance has been obtained for the work from the UHAS research and Ethics committee (Protocol Identification Number: UHAS-REC 4.7 l7l2t-22) and from the University of Ghana Medical and Research Centre (Protocol Identification Number: UGMC-IRB/MSRC/011/2022).

## 5. Conclusions

From our study, the incidence rates of COVID-19 among the pediatric population admitted at the COVID-19 treatment center of the UGMC in Ghana were reported to be higher in 2021 than in 2020. They were also observed to be higher among the age group of 6 to 10 years and the male gender population.

The commonly reported symptoms among these patients included nasal congestion, fever and cough and were usually mild to moderate in severity. However, the mortality rate was higher among those children subsequently transferred to the ICU.

Azithromycin and hydroxychloroquine were the most prescribed combination of antimicrobial agents used in the management of COVID-19 among the pediatric patients in our study. This reflected the recommendations in the current Ghanian STG; however, overall compliance with the current STG was sub-optimal. Knowledge of the clinical presentation as well as management and the impact of COVID-19 on pediatric patients is important to ensure that appropriate resources and skillful personnel are allocated for their care. In addition, there is an urgent need to revise the Ghana STG given concerns with over-promoting the prescribing of antimicrobials. We will be following this up in the future.

## Figures and Tables

**Figure 1 antibiotics-12-00283-f001:**
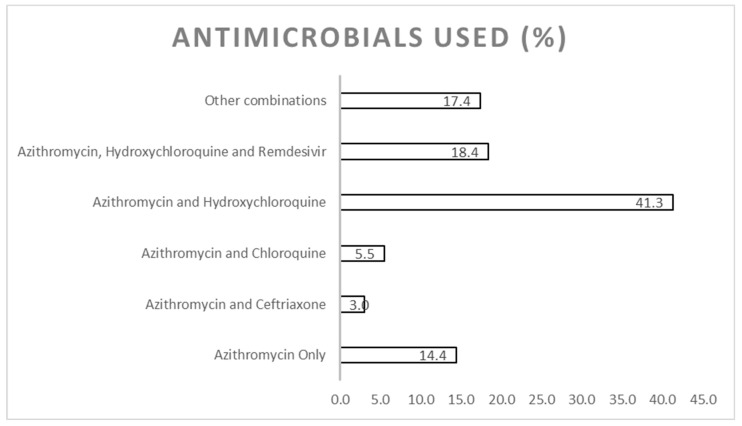
Antimicrobial agents prescribed for the management of COVID-19 infection among pediatric patients at the University of Ghana Medical Centre.

**Table 1 antibiotics-12-00283-t001:** Association between compliance to the COVID-19 Standard Treatment Guidelines (STG) and patients’ characteristics.

Characteristics (*n*)	Total, *n* (%)	Compliance to STG		Chi Square *p*-Value
		Yes, *n* (%)	No, *n* (%)	
Age Category (*n* = 201)				0.405
Less than 5	41 (20.4%)	32 (78.0%)	9 (22.0%)	
6–10 years	90 (44.8%)	59 (65.6%)	31(34.4%)	
11–15 years	56 (27.9%)	38 (67.9%)	18 (32.1%)	
16–18 years	14 (6.9%)	8 (57.1%)	6 (42.9%)	
Year of Admission (*n* = 201)				0.368
2020	85 (42.3%)	55 (64.7%)	30(35.3%)	
2021	116 (57.7)	82 (70.7%)	34(29.3%)	
Gender (*n* = 201)				0.213
Male	104 (51.7%)	75 (72.1%)	29(27.9%)	
Female	97 (48.3%)	62 (63.9%)	35(36.1%)	
Complications present (*n* = 201)				0.462
Acute Kidney Injury	19 (9.5%)	15 (78.9%)	4 (21.1%)	
Acute Liver Disease	11 (5.5%)	8 (72.7%)	3 (27.3%)	
COVID-Pneumonia	162 (80.6%)	109(67.3%)	53(32.7%)	
Septic Shock	9 (4.4%)	5 (55.6%)	4 (44.4%)	
Presence of co-infection diagnosed (*n* = 201)				0.639
Yes	61 (30.4)	18 (29.5)	46 (32.9)	
No	140 (69.6)	43 (70.5)	94 (67.1)	
Type of co-infection diagnosed (*n* = 201)				0.533
None	140 (69.6)	94 (67.1)	36 (25.7)	
Bacterial co-infection	30 (14.9)	23 (76.7)	7 (23.3)	
Other co-infections	31 (15.5)	20 (64.5)	11 (35.5)	
Level of Severity (*n* = 201)				0.988
Mild/Moderate	77 (38.3%)	52 (67.5%)	25(32.5%)	
Severe Pneumonia	64 (31.8%)	44 (68.8%)	20(31.3%)	
Severe Pneumonia with ARDS/Sepsis	60 (29.9%)	41 (68.3%)	19(31.7%)	
Cough as presenting complaint (*n* = 201)				0.623
Yes	32 (15.9%)	23 (7 1.9%)	9 (28.1%)	
No	169 (84.1%)	114(67.5%)	55(32.5%)	
Sore Throat as presenting complaint (*n* = 201)				**0.026**
Yes	23 (11.4%)	11 (47.8%)	12(52.2%)	
No	178 (88.6%)	126(70.8%)	52(29.2%)	
Fever > 37 °C as a presenting complaint (*n* = 201)				0.898
Yes	191 (95.0%)	130(68.1%)	61(31.9%)	
No	10 (5.0%)	7 (70.0%)	3 (30.0%)	
Presenting complaint of nasal congestion (*n* = 201)				0.057
Yes	193 (96.0%)	134(69.4%)	59(30.6%)	
No	8 (4.0%)	3 (37.5%)	5 (62.5%)	
Method of radiological assessment used in diagnosis (*n* = 201)				**0.044**
Chest CT	27 (13.4%)	22 (81.5%)	5 (18.5%)	
Chest X-ray	44 (21.9%)	30 (68.2%)	14(31.8%)	
Both	31 (15.4%)	15 (48.4%)	16(51.6%)	
None	99 (49.3%)	70 (70.7%)	29 (29.3%)	
Treatment Outcome (*n* = 201)				0.331
Discharge home	179 (89.1%)	120 (67.0%)	59 (33.0%)	
Mortality	22 (10.9%)	17 (77.3%)	5 (22.7%)	
Transferred to ICU (*n* = 201)				0.623
Yes	19 (9.5%)	12 (63.2%)	7 (36.8%)	
No	182 (90.5%)	125(68.7%)	57 (31.3%)	
Duration of Admission (*n* = 201)				0.243
Less than 7 days	131 (65.2%)	84 (64.1%)	47 (35.9%)	
8–14 days	66 (32.8%)	50 (75.8%)	16 (24.2%)	
15–21 days	4 (2.0%)	3 (75.0%)	1 (25.0%)	

NB: Variables with *p*-values highlighted in bold showed a statistically significant association with compliance to the COVID-19 Standard Treatment Guidelines.

**Table 2 antibiotics-12-00283-t002:** Association between treatment outcomes and patients’ characteristics.

Characteristics (*n*)	Total, *n* (%)	Treatment Outcome		Chi Square *p*-Value
		Discharge Home, *n* (%)	Death, *n* (%)	
Age Category (*n* = 201)				**0.020**
Less than 5	41 (20.4%)	37(90.2%)	4 (9.8%)	
6–10 years	90 (44.8%)	81 (90.0%)	9 (10.0%)	
11–15 years	56 (27.9%)	52 (92.9%)	4 (7.1%)	
16–18 years	14 (6.9%)	9 (64.3%)	5 (35.7%)	
Gender (*n* = 201)				0.780
Male	104 (51.7%)	92 (88.5%)	12 (11.5%)	
Female	97 (48.3%)	87 (89.7%)	10 (10.3%)	
Year of Admission (*n* = 201)				0.292
2020	85 (42.3%)	78 (91.8%)	7 (8.2%)	
2021	116 (57.7)	101 (87.1%)	15 (12.9%)	
Complications present (*n* = 201)				0.581
Acute Kidney Injury	19 (9.5%)	17 (89.5%)	2 (10.5%)	
Acute Liver Disease	11 (5.5%)	8 (72.7%)	3 (27.3%)	
COVID-Pneumonia	162 (80.6%)	146 (90.1%)	16 (9.9%)	
Septic Shock	9 (4.4%)	8 (88.9%)	1 (11.1%)	
Level of Severity (*n* = 201)				0.523
Mild/Moderate	77 (38.3%)	71 (92.2%)	6 (7.8%)	
Severe Pneumonia	64 (31.8%)	56 (87.5%)	8 (12.5%)	
Severe Pneumonia with ARDS/Sepsis	60 (29.9%)	52 (86.7%)	8 (13.3%)	
Method of radiological assessment used in diagnosis (*n* = 201)				0.322
Chest CT	27 (13.4%)	24 (88.9%)	3 (13.7%)	
Chest X-ray	44 (21.9%)	37 (84.1%)	7 (15.9%)	
Both	31 (15.4%)	26 (83.9%)	5 (16.1%)	
None	99 (49.3%)	92 (92.9%)	7 (7.1%)	
Presence of co-infection diagnosed (*n* = 201)				0.188
Yes	61 (30.4)	57 (93.4)	4 (6.6)	
No	140 (69.6)	122 (87.1)	18 (12.9)	
Type of co-infection diagnosed (*n* = 201)				0.421
None	140 (69.6)	122 (87.1)	18 (12.9)	
Bacterial coinfection	30 (14.9)	28 (93.3)	2 (6.7)	
Other co-infections	31 (15.5)	29 (93.5)	2 (6.5)	
Transferred to ICU (*n* = 201)				**0.002**
Yes	19 (9.5%)	12(63.2%)	7 (36.8%)	
No	182 (90.5%)	167 (91.8%)	15 (8.2%)	
Duration of Admission (*n* = 201)				**0.025**
Less than 7 days	131 (65.2%)	122 (93.1%)	9 (6.9%)	
8–14 days	66 (32.8%)	54 (81.8%)	12 (18.2%)	
15–21 days	4 (2%)	3 (75%)	1 (25%)	
Compliance to COVID STG (*n* = 201)				0.468
Yes	137 (68.2%)	120 (87.6%)	17 (12.4%)	
No	64 (31.8%)	59 (92.2%)	5 (7.8%)	

NB: Variables with *p*-values highlighted in bold showed a statistically significant association with Treatment outcomes.

**Table 3 antibiotics-12-00283-t003:** Multiple logistic regression between independent variables that showed statistically significant association with STG compliance.

Independent Characteristics	Adjusted Odds Ratio	95% CI	*p*-Value
Sore throat as presenting complaint	0.39	0.16–0.94	**0.036**
Method of radiological assessment used in diagnosis	1.21	0.93–1.57	0.160

NB: Variables with *p*-values highlighted in bold showed a statistically significant association with compliance to the COVID-19 Standard Treatment Guidelines.

**Table 4 antibiotics-12-00283-t004:** Logistic regression between independent variables that showed statistically significant association with treatment outcomes.

Independent Characteristics	Adjusted Odds Ratio	95% CI	*p*-Value
Age	0.69	0.41–1.19	0.190
Transferred to ICU	0.22	0.07–0.66	**0.007**
Duration of Admission	0.44	0.19–1.00	0.052

NB: Variables with *p*-values highlighted in bold showed statistically significant association with treatment outcome.

## Data Availability

Additional data are available from the corresponding authors on reasonable request.

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
