# Peer review of "Assessing the Clinical Characteristics and Management of COVID-19 among Pediatric Patients in Ghana: Findings and Implications"

_antibiotics, 2023, doi:10.3390/antibiotics12020283_

Round 1

Reviewer 1 Report

Sefah and colleagues present a study of 201 pediatric patients with hospital admission for lab-confirmed SARS-CoV-2 infection. The authors show that select clinical and demographic factors are associated with compliance with a local standard treatment guideline.  Furthermore, patient age and admission to the ICU were associated with increased odds of mortality.  Sub-optimal antimicrobial prescribing was also observed.  

Major Points:

More information regarding the STG as needed, particularly description on desired clinical decision making based on clinical presentation as well as treatment recommendations. Without that information, the reader may be unclear as to whether the STG outlines which symptom(s) to consider when the clinician decides on the treatment strategy or radiology assessments for a COVID patient.  Demonstrating the possibility that STG had the opportunity to be independent from symptoms and more importantly clinical resource utilization is essential for interpreting the statistical tests.

Table 1 shows cough was only 15.9% which seems quite low, especially since the authors specify respiratory symptoms as one of the most common presenting symptoms among admitted patients, both in the Background and Discussion.  If this proportion is correct, the authors should consider a brief discussion on why this symptom differs significantly from other studies.

Results presented in Table 1 and the Discussion suggests that all 201 patients had complications.  This seems to conflict with statements such as “161 were discharged home with no complications” [Line 135]. Are the remaining 18 patients discharged home assumed to have complications (i.e., 161 home w/o complications of 179 discharged home)?  Relatedly, it appears that all (or nearly all) patients received treatment for their COVID admission.  Having a cohort where all patients had complications and treatment is not a study flaw.  However, if true, I would recommend the authors modify the text when discussing the inclusion/exclusion criteria. Clarifications are needed either way. The reviewer would also suggest including the treatment denominator either in the text (lines 136-41 ) or in the Figure 1 title.

Radiology findings is a nominal variable, which make the OR interpretation in Table 3 unclear. The authors make the assumption of an incremental increase with the 2 ordinal variables in Table 4.  This assumption may not be valid given the sample size imbalance with categories. Refinement and validation of the regression models is recommended.

Minor Points:

The denominator changes from the 201 (Lines 134 and 227).

Presumably the event of interest in Table 4 is discharge home, but it’s unclear why mortality (rare outcome) was not selected.

Was the mortality rate or STG compliance for the 1787 (1988-201) COVID patients not included in the study different? These data may not be available. However, it would help demonstrate possible selection biases.

X-ray and CT had higher non-compliance with STG?  Is this tied to severity?

Author Response

Comments and Suggestions for Authors

Sefah and colleagues present a study of 201 pediatric patients with hospital admission for lab-confirmed SARS-CoV-2 infection. The authors show that select clinical and demographic factors are associated with compliance with a local standard treatment guideline.  Furthermore, patient age and admission to the ICU were associated with increased odds of mortality.  Sub-optimal antimicrobial prescribing was also observed.  

Author comments: Thank you for this summary

A) Major Points:

i) More information regarding the STG as needed, particularly description on desired clinical decision making based on clinical presentation as well as treatment recommendations. Without that information, the reader may be unclear as to whether the STG outlines which symptom(s) to consider when the clinician decides on the treatment strategy or radiology assessments for a COVID patient.  Demonstrating the possibility that STG had the opportunity to be independent from symptoms and more importantly clinical resource utilization is essential for interpreting the statistical tests.

Author comments: Thank you for this. We may now added extra information on the consideration of the Ghana STG for treatment strategies in the introduction section of the manuscript, and hope this is now acceptable

ii) Table 1 shows cough was only 15.9% which seems quite low, especially since the authors specify respiratory symptoms as one of the most common presenting symptoms among admitted patients, both in the Background and Discussion.  If this proportion is correct, the authors should consider a brief discussion on why this symptom differs significantly from other studies.

Author comments: Thank you for this feedback. We have added extra details to explain this deviation from previous studies. As the reviewer is aware, in these types of study we are totally reliant on the comments in patients’ notes without going back to the relevant physicians. We hope this is now acceptable.

iii) Results presented in Table 1 and the Discussion suggests that all 201 patients had complications.  This seems to conflict with statements such as “161 were discharged home with no complications” [Line 135]. Are the remaining 18 patients discharged home assumed to have complications (i.e., 161 home w/o complications of 179 discharged home)?  Relatedly, it appears that all (or nearly all) patients received treatment for their COVID admission.  Having a cohort where all patients had complications and treatment is not a study flaw.  However, if true, I would recommend the authors modify the text when discussing the inclusion/exclusion criteria. Clarifications are needed either way. The reviewer would also suggest including the treatment denominator either in the text (lines 136-41 ) or in the Figure 1 title.

Author comments: Thank you for this correction. We have corrected the conflicting statement captured in line 135 by stating that “the study recorded an overall mortality rate of 10.9% (22/201) among our study population. Our inclusion criteria was paediatric patients diagnosed COVID-19 through laboratory confirmation of SAR-CoV-2 infection which has been stated in the method section. The Ghana treatment guideline recommends the use of antimicrobials for all laboratory confirmed cases of COVID-19 with or without symptoms. These details have been stated in both the introduction and discussion sections of the paper. We hope this clarifies why all confirmed cases who happen to have one form of complication or another were all offered treatment.

iv) Radiology findings is a nominal variable, which make the OR interpretation in Table 3 unclear. The authors make the assumption of an incremental increase with the 2 ordinal variables in Table 4.  This assumption may not be valid given the sample size imbalance with categories. Refinement and validation of the regression models is recommended.

Author comments: Thank you for this observation. We are aware of the nominal nature of radiology variable. Logistic regression was conducted between the STG compliance (a binary variable) and the radiological findings (categorical variable). There was no associated between these two variables and therefore we were silent on its significant effect on the outcome. We hope this explanation clarifies our findings.

B) Minor Points:

i) The denominator changes from the 201 (Lines 134 and 227).

Author comments: Thank you for this. We have corrected this observation in both lines to reflect the correct denominator of 201, and hope this is now OK.

ii) Presumably the event of interest in Table 4 is discharge home, but it’s unclear why mortality (rare outcome) was not selected.

Author comments: Thank you for this. We made the “mortality” sub-category of treatment outcome as the reference variable to the “discharge home” variable and therefore our interpretation of the regression analysis. We hope this is now acceptable.

iii) Was the mortality rate or STG compliance for the 1787 (1988-201) COVID patients not included in the study different? These data may not be available. However, it would help demonstrate possible selection biases.

Author comments: Thank you for this. The 201 cases were COVID-19 diagnosed patients for the 2 two-year period. Our assessment of the mortality and STG compliance rates were for just the 201 COVID-19 cases. We hope this clarifies our findings.

iv) X-ray and CT had higher non-compliance with STG?  Is this tied to severity?

 Author comments: Thank you for this. Our finding rather shows the reverse with higher compliance rates observed for those with CT (81.5%) and X-ray diagnoses (68.2%). Our study showed no association between compliance and level of severity as captured in our results. We hope this explanation helps to clarify your observation.

Reviewer 2 Report

Congratulations, it is a very well written manuscript!

In my opinion there are too many keywords, too many references.

Reference 12 may be incomplete, the year of the appearance is missing.

Reference 67 also incomplete.

Author Response

Comments and Suggestions for Authors

Congratulations, it is a very well written manuscript!

Author comments: Thank you – greatly appreciated

In my opinion there are too many keywords, too many references.

Author comments: Thank you – we have now revised the paper including cutting down on the number of key words as well as reducing the number of references where we can, and hope it is now acceptable.

Reference 12 may be incomplete, the year of the appearance is missing.

Author comment: Thank you – you are correct. However, we have now removed this and a number of other references to cut these down where we can. We hope this is now acceptable. 

Reference 67 also incomplete.

Author comment: Thank you (now 51). This is the reference stated on the URL include the date, and we hope this is now OK.

Reviewer 3 Report

The authors provided insight to the clinical aspect and the management of pediatric patients infected with Covid-19 from a quaternary centre in Ghana. I appreciate the efforts and dedication about what your team have studied. However, I have several questions as well as some suggestions for a more cursive manuscript and a better understanding of the provided subject.

1.     The aim of the study could be presented with more clarity, especially in the abstract section of the manuscript. Does STG (line 38) signify the standard treatment guidelines?

2.     The manuscript does not present the subject in a standardized manner for an observational study. The Materials and Methods section should be placed between the Introduction and the Results section for a better understanding of the study. The authors should check the STROBE(Strengthening the Reporting of Observational Studies in Epidemiology) initiative internet page in order to modify the manuscript structure.

3.     The inclusion and exclusion criteria should be mentioned in the Materials and Methods section.

4.     What is the median age in the studied sample?

5.     “80.0% (161/201) of all admission were dis charged home with no untreated complications, with an overall mortality rate of 10.9%(22/201)” – lines 134 to 135 –  Is 10.9% the overall mortality rate concerning the entire sample?

6.     How did the authors define disease severity? Did they use a standardized classification or protocol? Did they use a severity score? It should be mentioned in the manuscript.

7.     The transfer to the ICU could also represent an outcome in the studied sample. Was there an association between the prescribed treatment and this aforementioned transfer?

8.     What was the national standard of care during the time the study took place? How did it change during this period of time? It should be mentioned in the text. Also, how did the authors quantify the compliance? Was it in complete accordance with the standard treatment guidelines or was it only partial?

9.     What was the indication for administering Azithromycin? Was it prescribed solely as an immunomodulating agent? Did the authors take into consideration the presence of superimposed infections when administering antimicrobials? Did the patients undergo any type of bacterial screening? Were some antimicrobials prescribed in an empirical manner? Also, were the inflammation markers involved in choosing the prescribed treatment? All of these issues should be noted and explained in the manuscript.

10.  After the initiation of the Covid-19 vaccination programme in Ghana, in march 2021, were any of the admitted patients already vaccinated?

11.  ‘’Compliance to the STG was found to be associated with a sore throat as a presenting complaint (p-value = 0.026) and the use of radiological findings for diagnosis (p-value = 0.05) (Table 1)’’ – lines 167-169. What do the authors imply by ’’ the use of radiological findings’’? Does it signify the presence of pulmonary infiltrates/ viral pneumonia?  How does it influence the compliance? Why did you choose these two variables?

There should be stated in the text how does the presence of a sore throat influences the compliance to the treatment guidelines. I’ve noticed that it is explained shortly in the Discussions section, but it should be detailed in the Results section as well, otherwise the association is not understood.

12.  “Treatment outcomes were also found to be associated with age category of the patients (p-value =0.048), the duration of their admission (p-value =0.025) and whether they were transferred to ICU (p-value =0.002)” – lines 170-172. The p-value concerning the age category presented in the table 2 (0.020) does not match the one in the text. Which one is the correct value?

13.  ‘’Treatment outcomes were independently predicted by the patients’ transfer to the ICU (aOR=0.22, CI = 0.07-0.66, p-value=0.007) (Table 4)’’ – lines 177-178. What is the treatment outcome stated in the sentence and table 4? This should be mentioned more clearly.

Lines 192 and 197 – Logistic regression and multiple logistic regression represent the same statistical analysis procedure. In the Materials and Methods section the paragraph concerning statistical analysis should explain better the tests that were used.

Thank you. 

Author Response

Comments and Suggestions for Authors

The authors provided insight to the clinical aspect and the management of pediatric patients infected with Covid-19 from a quaternary centre in Ghana. I appreciate the efforts and dedication about what your team have studied. However, I have several questions as well as some suggestions for a more cursive manuscript and a better understanding of the provided subject.

Author comments: Thank you for your positive comments, and we hope we have adequately addressed areas of concern.

  1. The aim of the study could be presented with more clarity, especially in the abstract section of the manuscript. Does STG (line 38) signify the standard treatment guidelines?

Author comments: Thank you for this. We have now clarified in our abstract that STG refers to the standard treatment guidelines for COVID-19 treatment and more clearly explained this in the introduction section.

  1. The manuscript does not present the subject in a standardized manner for an observational study. The Materials and Methods section should be placed between the Introduction and the Results section for a better understanding of the study. The authors should check the STROBE(Strengthening the Reporting of Observational Studies in Epidemiology) initiative internet page in order to modify the manuscript structure.

Author comments: Thank you for this. As you are aware – we have followed the layout of the paper as directed by Antibiotics. In addition, reported the findings in line with many other similar PPS studies we have undertaken (with the references documented). We hope this is acceptable.

  1. The inclusion and exclusion criteria should be mentioned in the Materials and Methods section.

Author comments: Thank you for this. The inclusion criteria for paediatric cases is mentioned in the “Study design, study site and population” portion of the method and material section. We hope this is now acceptable.

  1. What is the median age in the studied sample?

Author comments: Thank you for this. We have now added the median age of the sampled cases to the results under the patient characteristics section. 

  1. “80.0% (161/201) of all admission were dis charged home with no untreated complications, with an overall mortality rate of 10.9% (22/201)” – lines 134 to 135 –  Is 10.9% the overall mortality rate concerning the entire sample?

Author comments: Thank you for this. The 10.9% mortality rate was for our sampled 201 cases admitted in 2020 and 2021. We have rephrased that section to make is clearer.

  1. How did the authors define disease severity? Did they use a standardized classification or protocol? Did they use a severity score? It should be mentioned in the manuscript.

Author comments: Thank you for this. Disease severity was based on criteria captured in the Ghana STG spanning from mild symptoms to severe forms such as ARDS/ Septic shock. We have added statement in the data collection section on how severity was defined, and hope this is now OK.

  1. The transfer to the ICU could also represent an outcome in the studied sample. Was there an association between the prescribed treatment and this aforementioned transfer?

Author comments: Thank you for this. We did not see an association between treatment compliance and transfer to ICU but rather saw an association between treatment outcomes and transfer to ICU. These are captured in both Table 1 and 2. We hope this is now OK.

  1. What was the national standard of care during the time the study took place? How did it change during this period of time? It should be mentioned in the text. Also, how did the authors quantify the compliance? Was it in complete accordance with the standard treatment guidelines or was it only partial?

Author comments: Thank you for this. We have now added extra information to the introduction on the Ghana standard of care for COVID-19 patients.  Compliance was determined based on choice of antimicrobials prescribed according to the level of severity of COVID-19 recommended by the 2020 edition of the Ghana COVID-19 STG. This has been captured in the method section under the heading “data collection method and analysis”, and we hope this is now acceptable.

  1. What was the indication for administering Azithromycin? Was it prescribed solely as an immunomodulating agent? Did the authors take into consideration the presence of superimposed infections when administering antimicrobials? Did the patients undergo any type of bacterial screening? Were some antimicrobials prescribed in an empirical manner? Also, were the inflammation markers involved in choosing the prescribed treatment? All of these issues should be noted and explained in the manuscript.

Author comments: Thank you for this. Per the Ghana STG, azithromycin was empirically indicated for all mild to moderate cases of COVID-19 in addition to hydroxychloroquine. We have explained this in detailed in both in the introduction and the discussion sections, and hope this now clarifies the situation.

  1. After the initiation of the Covid-19 vaccination programme in Ghana, in march 2021, were any of the admitted patients already vaccinated?

Author comments: Thank you for. We have explained in our paper that children under 15 years are currently not given COVID-19 vaccination in Ghana. This concern has been elaborated in the discussion section, and hope this is now OK.

  1. ‘’Compliance to the STG was found to be associated with a sore throat as a presenting complaint (p-value = 0.026) and the use of radiological findings for diagnosis (p-value = 0.05) (Table 1)’’ – lines 167-169. What do the authors imply by ’’ the use of radiological findings’’? Does it signify the presence of pulmonary infiltrates/ viral pneumonia?  How does it influence the compliance? Why did you choose these two variables?

Author comments: Thank you for this. We have now rephrased the term “the use of radiological findings” to “Method of radiological assessment used in diagnosis” to make it clearer as it refers to the type of radiological assessment that was performed to support management. We did not collect information of the findings of the assessment and so we hope this amendment will help clarify the meaning of that variable.

  1. There should be stated in the text how does the presence of a sore throat influences the compliance to the treatment guidelines. I’ve noticed that it is explained shortly in the Discussions section, but it should be detailed in the Results section as well, otherwise the association is not understood.

Author comments: Thank you for this. We have now added extra information the association between compliance as sore throat as a symptom among our study population, and hope this is now acceptable.

  1. “Treatment outcomes were also found to be associated with age category of the patients (p-value =0.048), the duration of their admission (p-value =0.025) and whether they were transferred to ICU (p-value =0.002)” – lines 170-172. The p-value concerning the age category presented in the table 2 (0.020) does not match the one in the text. Which one is the correct value?

Author comments: Thank you for this. We have now corrected this value.

  1. ‘’Treatment outcomes were independently predicted by the patients’ transfer to the ICU (aOR=0.22, CI = 0.07-0.66, p-value=0.007) (Table 4)’’ – lines 177-178. What is the treatment outcome stated in the sentence and table 4? This should be mentioned more clearly.

Author comments: Thank you for this. The type of treatment outcomes assessed in this study have been captured in the method section and Table 1 in the result section, and hope this is now OK.

15) Lines 192 and 197 – Logistic regression and multiple logistic regression represent the same statistical analysis procedure. In the Materials and Methods section the paragraph concerning statistical analysis should explain better the tests that were used.

Author comments: Thank you for this. We have not stated clearly the type of multivariate analysis (i.e. multiple logistic regression), and hope this is now OK.

Round 2

Reviewer 3 Report

Israel Abebrese Sefah et al studied the characteristics and management of Covid-19 among pediatric patients and their relationship with treatment compliance.  The authors have improved the quality of the manuscript and provided answers to most of the issues raised. However, some subjects were not addressed clearly.

1.       “Author comments: Thank you for this. Per the Ghana STG, azithromycin was empirically indicated for all mild to moderate cases of COVID-19 in addition to hydroxychloroquine. We have explained this in detailed in both in the introduction and the discussion sections, and hope this now clarifies the situation.  This particular sentence should be stated in the manuscript, in the introduction section: ”azithromycin was empirically indicated for all mild to moderate cases of COVID-19 in addition to hydroxychloroquine”.

2.       The following questions were not addressed and I find them particularly significant concerning antibiotic stewardship.  “Did the authors take into consideration the presence of superimposed infections when administering antimicrobials? Did the patients undergo any type of bacterial screening? Were some antimicrobials prescribed in an empirical manner? Also, were the inflammation markers involved in choosing the prescribed treatment?” If a bacterial screening had not been performed it should be stated in the text.

3.       The study limitations should be noted in the last part of the discussion section

Author Response

Comments and Suggestions for Authors

Israel Abebrese Sefah et al studied the characteristics and management of Covid-19 among pediatric patients and their relationship with treatment compliance.  The authors have improved the quality of the manuscript and provided answers to most of the issues raised. However, some subjects were not addressed clearly.

Author comments: Thank you for this. We hope we have now adequately addressed these.

  1. “Author comments: Thank you for this. Per the Ghana STG, azithromycin was empirically indicated for all mild to moderate cases of COVID-19 in addition to hydroxychloroquine. We have explained this in detailed in both in the introduction and the discussion sections, and hope this now clarifies the situation.“   This particular sentence should be stated in the manuscript, in the introduction section: ”azithromycin was empirically indicated for all mild to moderate cases of COVID-19 in addition to hydroxychloroquine”.

Author comments: Thank you now added.

  1. The following questions were not addressed and I find them particularly significant concerning antibiotic stewardship.  “Did the authors take into consideration the presence of superimposed infections when administering antimicrobials? Did the patients undergo any type of bacterial screening? Were some antimicrobials prescribed in an empirical manner? Also, were the inflammation markers involved in choosing the prescribed treatment?” If a bacterial screening had not been performed it should be stated in the text.

Author comments: Thank you for this. We have added data on co-infection diagnosis as well as details of the co-infections and their association with compliance to COVID STG and treatment outcomes in the updated Results and Discussion sections. We did not collect data on the type of biomarkers used in the screening of the bacteria co-infection as a way of confirming the suspected of the bacteria co-infection – as this was a retrospective study based on the contents of patients’ notes -and have indicated this as limitation to our findings. We hope this is now acceptable.

  1. The study limitations should be noted in the last part of the discussion section

Author comments: Thank you – now added.
